# Recent Progress of Terahertz Spatial Light Modulators: Materials, Principles and Applications

**DOI:** 10.3390/mi13101637

**Published:** 2022-09-29

**Authors:** Shengnan Guan, Jierong Cheng, Shengjiang Chang

**Affiliations:** 1Institute of Modern Optics, Nankai University, No. 38 Tongyan Road, Tianjin 300350, China; 2Tianjin Key Laboratory of Micro-Scale Optical Information Science and Technology, Tianjin 300350, China; 3Tianjin Key Laboratory of Optoelectronic Sensor and Sensing Network Technology, Tianjin 300350, China

**Keywords:** terahertz, modulation, semiconductor, liquid crystal, graphene, VO_2_, modulation depth, modulation rate, single-pixel imaging, beam scanning

## Abstract

Terahertz (THz) technology offers unparalleled opportunities in a wide variety of applications, ranging from imaging and spectroscopy to communications and quality control, where lack of efficient modulation devices poses a major bottleneck. Spatial modulation allows for dynamically encoding various spatial information into the THz wavefront by electrical or optical control. It plays a key role in single-pixel imaging, beam scanning and wavefront shaping. Although mature techniques from the microwave and optical band are not readily applicable when scaled to the THz band, the rise of metasurfaces and the advance of new materials do inspire new possibilities. In this review, we summarize the recent progress of THz spatial light modulators from the perspective of functional materials and analyze their modulation principles, specifications, applications and possible challenges. We envision new advances of this technique in the near future to promote THz applications in different fields.

## 1. Introduction

Spatial light modulators (SLMs) [1] are two-dimensional planar devices for the modulation of beam intensity [2,3], phase [4,5], polarization or combination of them [6] in space and in real time. They are composed of arrays of pixels with individually controllable responses. Once a two-dimensional pattern is written into the device through electrical or optical methods, the passing beam will be encoded with a spatial function. As well-established technology, SLMs have wide applications in beam scanning [7], wavefront correction [8], imaging [9], information processing [10], spatial filtering [11] and so on.

In recent years, terahertz waves (0.1–10 THz) located between microwaves and infrared waves in the spectrum [12] have drawn worldwide attention due to potential applications in spectral detection, imaging and next-generation communications [13], over which SLMs play a vital role. However, the responses of natural materials to THz waves are very weak, resulting in the lack of effective THz SLMs.

The spatial modulation of THz waves was initially performed by mechanically changing hard masks engraved in metallic plates with different patterns [14]. The on/off modulation shows good contrast over wide bandwidth. However, the modulation parameters are not reconfigurable, and the modulation speed is limited. The repeated optical alignment leads to poor system stability [15,16]. SLMs based on liquid crystals (LC) [17,18] and digital micromirror devices (DMD) [19] are widely used in optical frequencies but encounter significant challenge in the THz region. The orientation of anisotropic LC molecules is electrically tilted for amplitude [20] or phase modulation [21,22]. The thickness of the LC should be increased dramatically for apparent modulation of THz waves due to their longer wavelength. The corresponding losses and applied voltages would be too high to be used. DMD is a micro-electromechanical system composed of thousands of miniature mirrors and the control circuits [23]. The micro-scale mirror is tilted under the electrostatic force, so that the reflected light will enter into or deviate from the detection direction for spatial intensity modulation [24]. DMDs in the THz band require much larger mirror size, which cannot be realized by existing manufacturing technology and pose more challenges to the control circuit. In addition, the PIN diodes [25,26] and varactors [27] commonly used in the microwave band will no longer be applicable due to the high loss. So, the development of THz SLMs cannot simply rescale the mature components from the nearby bands. 

The rise of metasurfaces [28] has brought a method to enhance the interaction between THz wave and tunable functional materials, such as semiconductors, phase change materials, LC, two-dimensional materials and nonlinear crystals [29,30]. The requirement for the external stimuli of electric field, magnetic field, optical pumping or temperature control, as a result, can be greatly relaxed. The two-dimensional array configuration of the metasurface naturally meets the needs of spatial modulation. Its subwavelength unit cells lead to high spatial resolution and large modulation depth, while it often narrows the working bandwidth due to the resonances. In addition, a SLM is much more complicated than a spatially uniform modulator as the former has to deal with additional problems of array integration, pixel control, crosstalk suppression, power management, heat dissipation, and so on. Researchers are actively searching for solutions of these problems to meet the requirement of THz applications. 

Although most THz SLMs are still in the laboratory stage, they have shown rich application prospects. For example, THz SLMs are used for single pixel imaging [31], which avoids raster scanning and expensive THz focal plane array detector. High resolution imaging is possible using the low-cost continuous-wave source and point detector. Combined with compressed sensing technology, THz super-resolution imaging has been realized in real time and even with video rate [32]. By coding appropriate spatial phase pattern, SLMs realize dynamic scanning of THz beam [33] and reconfigurable control of its radar cross section [34], which is of great significance in military and communication applications. THz SLMs are used for dynamic holographic imaging [35] and generation of THz vortex beams [36,37], which greatly enrich the means of beam manipulation.

In recent years, many comprehensive reviews have emerged in this research area. Overviews of reconfigurable metamaterials were provided for spectrally or spatially manipulating the intensity, phase and polarization of terahertz waves [38]. The progress of THz modulators has been summarized in the terahertz band with the main focus on spatially uniform temporal modulation [39,40,41,42]. Some other reviews pay more attention to applications in single pixel imaging [31]. However, there are few reports emphasizing the spatial modulators themselves with pixel-level control. In this paper, we comprehensively discuss the research progress of THz SLMs as classified by tunable functional materials as shown in Figure 1. Amplitude and phase modulation under different type of stimuli are fully covered. Structures, mechanism and specifications are detailed. Technical challenges and application prospects are discussed. 

This paper is organized as follows. In Section 2, Section 3, Section 4, Section 5 and Section 6, we introduce different types of THz SLMs using mechanical systems, liquid crystals, semiconductors, vanadium dioxide, and graphene based on electrical, optical, thermal and mechanical stimuli. In Section 7, we describe the terahertz spatial modulation during generation and detection of the THz beam. In Section 8, the comparison and analysis of the pros and cons of different modulators is conducted in terms of modulation depth, modulation speed, working bandwidth, spatial resolution, pixel number and so on. In Section 9, the advanced application scenarios are summarized, and technical challenges are discussed. Finally, a brief summary of this rapidly expanding field is provided in Section 10.

## 2. Mechanically Tunable SLMs

This section summarizes the SLMs working in the THz band whose elements are mechanically deformable driven by external electrical or thermal stimuli. Additionally, the deformation leads to the redirection of the beam or modulation of the local resonances. 

DMD is a mature spatial light modulator in visible and infrared bands composed of millions of micromirrors [19]. The size of the mirror is around 10 μm. The orientation of each mirror is independently controlled by the electrostatic force, so as to reflect the light to the detector direction or other directions, and to switch the pixel between 0 and 1 states. The wavelength of THz wave is in the order of tens of and hundreds of micrometers, and the corresponding mirror size should be in the order of millimeters. This large mirror array is difficult in fabrication. Even if possible, the mirror cannot be effectively tilted as the required driving voltage is too high. 

A straightforward solution is to combine multiple micromirrors in an optical DMD as a THz pixel [43]. In state 0, the pixel is a flat mirror for specular reflection. In state 1, all micromirrors in a pixel are inclined, forming a blazed grating in the Littrow condition, and the reflected wave returns back into the original path. The structure has the advantages of broad bandwidth and large modulation depth of more than 60% in the range of 1.7–3 THz. Here, the modulation depth is defined as
(1)$$\mathrm{modulation}\;\mathrm{depth}=\left|I_{ON}-I_{OFF}\right|/\max\left(I_{ON},\;I_{OFF}\right)$$
which is applicable throughout the paper. The pixel size can be changed dynamically by grouping different number of micromirrors according to the demand. There is a balanced pixel size that could ensure high spatial resolution and reduce the edge diffraction.

In addition, the micromirrors can be driven by electrostatic forces. A self-assembled THz DMD scheme [44] was proposed in Figure 2. The residual stress of the metal structure makes the micromirror naturally tilt up, and the electrostatic force pulls the micromirror towards the substrate. So, the tilted-up structure diffracts the THz wave to non-specular directions, while the flat structure directs the beam to the detector. Switching between the two states is realized through 37 V voltage. The size of the micromirror is 220 × 100 μm^2^. A THz SLM is composed of 24 pixels with each pixel composed of 4 × 8 micromirrors. Over the wide range of 0.97–2.28 THz, the study shows modulation depth of more than 50% and modulation speed up to 1 kHz. However, the inclination angle of the micromirror in each pixel cannot be accurately controlled and may have some nonuniformity [44]. Many subwavelength metallic cantilevers are electrically actuated for the generation of different grating patterns. The applications for beam steering exceed 40° and beam focusing have been demonstrated [45].

The micro-electro-mechanical system is further combined with metasurfaces for resonance modulation. In 2009, Hu Tao et al. demonstrated a reconfigurable metamaterial by connecting the split ring resonators with two cantilever legs. As the cantilever bends in response to the external stimuli, the split rings are driven to tilt out of the plane to dramatically change the electromagnetic responses [46]. In 2014, Zhengli Han et al. embedded a cantilever disk in the split ring resonator unit cell, which forms an electrostatically tunable capacitor for switching on or off the resonance with high contrast of 16.5 dB at 480 GHz [47]. In 2021, Ruijia Xu et al. designed multilayer cantilevers which naturally bend upward due to the mismatch of the thermal expansion coefficients. The deformation of the cantilevers is actuated by the DC bias voltage for resonance intensity modulation at 0.33 THz [48]. Due to the fabrication complexity, the cantilevers in a metasurface are usually fed by a single driving voltage and actuated uniformly in space. The pixel-level electromechanical control of the micro resonators is still a challenge. 

## 3. LC SLMs

LC is a type of polymer with crystalline lattice and fluidity. The orientation of the elongated LC molecules changes with the applied electric field to offer tunable refractive index. So, LC is widely used in SLMs especially for spatial phase modulation based on the anisotropy. When applied in the THz band, the birefringence of LC becomes smaller than that in the optical band, and the required thickness of the LC layer is often more than 100 μm due to the longer wavelength, which leads to a slow modulation speed, high driving voltage and strong absorption [49]. 

Metasurfaces composed of a metallic pattern-insulator-metal film have been demonstrated for perfect absorption with ultrathin thickness down to λ/10 [50]. By inserting LC into the ultrathin insulator layer, the metasurface becomes an active yet efficient absorber for amplitude or phase modulation. Next, we discuss the LC metasurface SLMs working in the reflection mode, the transmission mode and both reflection and transmission modes.

Willie J. Padilla et al. filled the isothiocyanate-LC mixture into the metasurface absorber [51] and designed a 6 × 6-pixel THz SLM in Figure 3a. The thickness of the LC layer is only 3 μm. Using the driving voltage of 15 V with 1 kHz repetition rate, the position of absorption peak is shifted and the overall amplitude modulation depth is as high as 75% at 3.67 THz. Recently, a programmable LC metasurface for THz beam steering is proposed by inserting NJU-LDN-4 LC into the insulating layer of the metallic pattern-insulator-metal metasurface [52]. A programmable linear array containing 24 pixels is fabricated. By applying a 40 V voltage, the reflected beam shows the opposite phase with constant amplitude. When the metasurface is encoded with a specific sequence of 40 V and 0 V, the reflected THz wave is bent by 32° due to the spatial phase gradient with reflection efficiency of 19.1%.

For spatial modulation in the transmission mode, one has to find a layered thin metasurface to incorporate LC for efficient beam manipulation. In Figure 3b, LC is filled into the complementary double-layer metal split ring metasurface [53]. The asymmetric split ring excites Fano resonance and ensures high transmittance near the resonance. By controlling the bias voltage as 0 or 10 V, the LC molecules are oriented along or perpendicular to the direction of the metal plate for binary phase modulation of 0° and 180°. The SLM is a 48 × 48 array with each two columns of elements being a pixel. The pixel size is 0.64 × 15.36 mm^2^. A 30 μm gap between adjacent pixels is used to block crosstalk. By electrically encoding different phase patterns, the metasurface is validated for generation of double beams, multiple beams and beams with arbitrary directions. 

When sandwiching LC with two layers of metasurfaces, the elements not only serve for resonant response but also function as electrodes of biasing [54], leading to compact LC cells. The response time of spatial phase modulation is tens of microseconds. With one layer of the metasurface being the interdigital electrode, the polarization sensitivity enables dual-mode modulation in reflection and transmission side for orthogonally polarized THz beam [55]. Based on this principle, an 8 × 8-pixel array realizes transmissive and reflective spatial modulation at two frequencies, respectively, through 60 V bias voltage with modulation depth of 38.8% and 66.1% in Figure 3c.

The introduction of metasurface reduces the thickness of the LC layer from hundreds of microns to tens of microns or even a few microns, improves the compactness of devices, increases the uniformity of the LC molecules and reduces the control voltage, but also limits the modulation bandwidth of the THz wave. Most of the metasurface structures behave as electrodes of bias circuit. Therefore, the orientation of LC molecules is still non-uniform within a pixel. Metasurface units with large filling factor in the lattice are preferable to improve the molecule uniformity. The division of pixels mainly depends on the isolated electrodes, that is, the connection and disconnection of metasurface units, while there is a lack of physical gap in the LC layer. It is often necessary to introduce anchor layers for initial alignment of LC molecules [55]. The pixel size is determined by the size of metasurface units, so LC SLMs often have high spatial resolution. For amplitude modulation, the modulation depth is not high due to the limited shift of the resonant frequency. For phase modulation, binary phase is often used instead of continuous phase to get rid of the amplitude variation. However, unwanted diffraction cannot be well suppressed using 1-bit coding. Quasi 2-bit phase coding is demonstrated in a LC metasurface by introducing an interdigital electrode. The enriched coding diversity is used for single-beam scanning up to ±20° [56]. In addition, due to the radiation loss of the metasurfaces, the efficiency of LC SLMs working in reflection or transmission mode needs to be improved, and the modulation rate is less than 1 kHz [51]. By selecting other types of LC, such as dual-frequency LC [57], the modulation rate is expected to be further improved. A recent study combines the dual-frequency LC with metasurface absorbers for spatial modulation with complementary patterns at two frequencies. Hadamard masks are achieved by frequency switching for single-pixel compressive imaging [58]. 

## 4. Semiconductor SLMs

Semiconductors, such as silicon (Si) [59] and gallium arsenide (GaAs) [60], with charge carrier dynamics under optical pump or electrical excitation are good candidates for THz modulation. The modulation schemes can be summarized as the injection or depletion of free charge carriers, whose concentration *N* determines the conductivity of a thin layer of semiconductors by the Drude model:(2)$$\sigma =\frac{Ne^{2}}{m\left(\omega +i/\tau \right)}$$
where *e* is the electron charge, *m* is the effective mass, and τ is the scattering time. The dynamic variation of the conductivity modulates the attenuation of the THz beam. We next separately review the progress of semiconductor SLMs based on electrical biasing and optical pumping.

### 4.1. Semiconductor SLMs with Electrical Biasing

Electronically controlled semiconductor SLMs use the Schottky contact to control the depletion and injection of carriers through voltage. A landmark study was conducted by Padilla where metal split ring resonator arrays are combined with n-GaAs to form Schottky junctions for resonance modulation [60]. When not biased, the carrier concentration of GaAs in the order of 10^16^ cm^−3^ shortens the split ring. The carriers are exhausted with a biased voltage and a strong resonance is formed. A 50% modulation depth at 0.72 THz is achieved when 16 V bias voltage is applied. The structure is further made into a 4 × 4 array, with each pixel controlled by a circuit [61], as shown in Figure 4a. The crosstalk noise between pixels is less than −30 dB. However, due to the large capacitance, the modulation rate of this structure is a few kHz. 

In order to improve the modulation rate of electrically controlled semiconductor SLMs, high electron mobility transistors are used to replace Schottky junction. The modulation rate can be increased to 10 MHz and the bias voltage can be further reduced [62]. In 2016, Saroj rout et al. prepared 2 × 2 pixel resonator array on top of the transistors [63]. The high-speed modulation of the THz amplitude is realized by controlling the channel carrier concentration with a low voltage of 1 V and a power consumption less than 1 mW. The modulation depth at 0.45 THz is 36%.

Another way to boost the modulation rate is to use the ultrathin metallic pattern-insulator-metal structure for reflective spatial light modulation [64]. An n-GaAs epitaxial layer is nestled between the metal microstructure and the ground plane. The carrier concentration of the epitaxial layer is changed to adjust the position and intensity of the absorption peak. Since the thickness of the epitaxial GaAs layer is only 2 μm, the equivalent capacitance is significantly reduced, and the modulation rate of the device is as high as 12 MHz. The average modulation depth is 62% over 26.5 V voltage control. This design achieves good balance between the modulation rate and the modulation depth. The metallic pattern-insulator-metal structure was further applied to single-pixel imaging. As shown in Figure 4b, the structure is made into an 8 × 8 array with 324 absorbing units per pixel [32]. The high-speed dynamic spatial coding is used for THz real-time imaging with 1 frame/s.

### 4.2. Semiconductor SLMs with Optical Pumping 

When the photon energy of the pump laser is greater than the semiconductor band gap, photoexcited carriers increase with the pump laser power if the diffusion is ignored [65]. The semiconductor has an increased absorption coefficient and the THz wave attenuates over a broad bandwidth [66]. The spatial modulation of the pump laser through commercialized DMD transfers the coding pattern to the semiconductor for THz modulation, with high spatial resolution limited by the optical diffraction limit.

In 2013, Padilla et al. used a DMD-patterned 980 nm continuous laser to irradiate high-resistivity silicon, resulting in different THz reflectivity in the irradiated area and the non-irradiated area [65]. They experimentally demonstrated a 31 × 33 pixel array with a pixel size of 328 × 328 μm^2^. The modulation depth is measured to be 43% at 0.7 THz when the pump laser power is increased to 1 W/cm^2^. Limited by the switching time of DMD, the overall modulation rate of the system is about 31 Hz.

The modulation effects on the reflected, transmitted and total internal reflected beams are compared in the optically pumped silicon, and the total internal reflected mode shows lower insertion loss and higher modulation depth [67], as shown in Figure 5a. The modulation depth reaches about 50% at the pump power of 500 mW/cm^2^. By taking the carrier response time into the consideration of the post-processing algorithm, the modulation rate can reach 20 kHz. The SLM is successfully used for real-time THz imaging and even a THz video with six frames per second.

Another way to improve the modulation depth is introduction of microscale or nanoscale structures at the interface of the semiconductor and air to reduce the reflection and to enhance the localization of the pump light. Some typical practice includes adding gold particles [68] and etching micro-pyramids on the high-resistivity silicon wafer as an optical antireflection layer [69]. As shown in Figure 5b, the reflectivity of the pump laser is reduced to 10%. Using 808 nm pump laser with power of 1 W, the modulation depth reaches 93.8%.

In addition, the above wafer-based SLM has relatively large thickness. When applied in the imaging system, the SLM is in the far field region of the object to be imaged, and the evanescent wave information cannot be modulated, and the modulation pattern blurs significantly after traveling through the wafer. By reducing the thickness of the semiconductor wafer to make it close to the object, super-resolution imaging can be realized. Therefore, near-field THz SLMs have been developed. Stanchev et al. used 150 μm [61] and 6 μm [71] silicon wafer as the SLM, with one side modulated by patterned femtosecond pulse and the other side attached to the object to be imaged. As shown in Figure 5c, the reduction in wafer thickness greatly reduces the degree of blurring of spatial mask. Using the femtosecond (fs) pump-terahertz detection system, the THz signal is detected within 5 ps after the femtosecond pulse to inhibit carrier diffusion. The modulation depth of 90% is achieved with fs fluence of 100 μJ/cm^2^. The single-pixel imaging has been realized successively with λ/4 and λ/45 imaging resolution. The carrier lifetime is in the order of sub nanoseconds, indicating a potentially high modulation rate.

## 5. Vanadium Dioxide (VO_2_) SLMs

Continuously reducing the thickness of semiconductor will lead to higher spatial resolution at the cost of reduced modulation depth [72]. VO_2_ is a metal oxide with a few hundred nanometer thickness. Its insulator-metal phase transition changes the conductivity by five orders of magnitude, making it an excellent material for THz modulation [73,74]. When the temperature exceeds 68 °C, VO_2_ changes from the insulating state to the metal state with the conductivity up to 4 × 10^5^ S/m, and the phase transition is repeatable. Electric or optical excitation is often used to change the lattice structure of VO_2_ through Joule heat induced temperature rise or carrier injection [75], so as to realize active SLM with high modulation depth.

VO_2_ thin films on both sides of sapphire substrate driven by electric current are prepared into a 2 × 2 array with pixel size of 4 × 4 mm^2^ [76]. As shown in Figure 6a, the modulation depth is greater than 95% over 0.13–0.9 THz. Gaps of 0.7 mm are kept between adjacent pixels, so that the crosstalk is less than −30 dB. However, the power consumption will increase severely as the number of pixels becomes large. In Figure 6b, Zhu group pumps the 180 nm thick VO_2_ film using a DMD encoded 800 nm fs pulse [77] with fluence of 14 mJ/cm^2^. VO_2_ changes from the insulating state to the metal state 50 ps after the femtosecond excitation, and the overall modulation depth is 30%. THz super-resolution imaging with resolution of λ/133 is realized by near-field modulation and compressed sensing technology.

Recently, Jin et al. proposed a nonvolatile VO_2_ programmable metasurface based on the hysteresis feature [78]. It consists of 8 × 8 pixels with each pixel composed of VO_2_ and the bowtie-shaped metallic structure. The electrically generated Joule heat changes the conductivity of VO_2_ and controls the reflection intensity of the THz wave. The modulation rate is 1 kHz. Due to the hysteresis property of VO_2_, each pixel can be electrically modulated in series. The image can be stored for 5 h at a constant temperature of 56 °C. This series modulation may greatly reduce the complexity of the control circuit. 

## 6. Graphene SLMs

Graphene is a two-dimensional semiconductor with zero band gap and extremely high carrier mobility. Its carrier concentration can be changed by chemical doping [79] or external bias voltage [80]. As compared to VO_2_, graphene shows a smaller modulation depth partly due to the one-atom thickness, and partly limited by the breakdown of the dielectric layer. As shown in Figure 7a, graphene is divided into 4 × 4 pixels by oxygen plasma showing the modulation depth of 50% as the voltage is tuned from −10 V to 40 V [81].

In 2015, Coskun Kocabas proposed an ionic liquid electrolyte gating method to tune the carrier concentration in a large dynamic range [82]. The ultra-thin electric double layer (EDL) formed at the graphene-electrolyte interface generates a large electric field without electric breakdown, which increases the charge density by a hundred times and improves the modulation depth. A transmissive graphene SLM is shown in Figure 7b. The graphene film on both sides of the ionic liquid electrolyte is divided into five 8 mm wide strips, and the directions of the graphene stripes in the two layers are orthogonal. Graphene is the tunable material and also the electrode. The intersection of each transverse and longitudinal electrode is defined as a pixel. Correspondingly, the voltage difference between the top and bottom electrodes of each pixel is used to modulate the transmission intensity of the THz beam. By applying a voltage difference of 2 V, THz transmittance is reduced from 65% to 32%. This structure is further developed into a 256 pixel SLM [84]. The transmittance modulation depth at 1 THz is as high as 80%, and the modulation depth can be maintained at 10% when the modulation rate is 1 kHz. This kind of SLM has a wide bandwidth, but the capacitance between the double-layer graphene limits further improvement of the modulation rate. Graphene stripes confine the carriers only in one direction and lead to obvious diffusion, so the resolution for imaging is limited.

Dynamic phase modulation accompanied with the carrier concentration variation is studied in graphene arrays [85]. However, the amplitude is inevitably varied with the phase, and the intensity of the THz wave is nonuniform. This is a common problem of THz SLMs made of semiconductors, metal oxides, graphene and other materials. It is not easy to modulate either amplitude or phase while maintaining the other. To keep the amplitude fixed, graphene metamaterials use 1 bit digital coding to reverse the reflected phase at two bias voltages [83]. As shown in Figure 7c, a certain number of graphene units are combined into one pixel, and FPGA is used to configure independent bias voltage for each pixel. Through the real-time regulation of the 0/1 state of the pixel, the dynamic radiation of single beam, double beam, multiple beams and diffusive reflection in the range of 1–1.9 THz are realized. The graphene metasurface with 1 bit phase modulation is further demonstrated for dynamic beam scanning from −5° to 20° at 0.98 THz through appropriate bias distribution [33].

The high carrier mobility of electrons and holes gives graphene broadband electromagnetic response. The atomic thickness makes it possible for near field modulation. Until now, preparation of the ultra-thin, large-area and uniform graphene film still faces challenges [86]. The electrically controlled independent pixels are mostly 1D structures, while the two-dimensional modulation is more complex and less studied.

## 7. THz Spatial Modulation during Generation

The above-mentioned THz SLMs reshape the beam pattern in the propagation process. In fact, spatial modulation can be combined with the generation or detection of the THz wave so that there is no need to introduce extra modulation components. Driven by the fs laser pulses around the near infrared or higher frequencies, THz radiation can be generated via optical rectification in nonlinear crystals or via spin-to-charge current conversion in ferromagnetic heterostructures. We next discuss the THz spatial modulation during the two types of generation. 

Nonlinear crystals with large second-order susceptibility include ZnTe [87], GaP [88], LiNbO_3_ [89] and DAST [90,91]. The THz electric field is determined by the polarization current I(t) as
(3)ETHz(t)∝χ(2)∂2I(t)∂t2
where $\chi^{\left(2\right)}$ is the second-order nonlinear susceptibility. Therefore, the spatial distribution of the THz wave can be directly controlled by shaping the pump laser. The modulation depth is related to the pump power. This configuration is similar to the optically pumped THz SLMs made of semiconductors. They both transfer the optical pattern into the THz pattern, but with drastically different schemes: optical rectification and photo-induced carriers, respectively. In 2018, Luana Olivieri et al. irradiated ZnTe crystal with DMD-encoded fs laser to produce structured THz radiation for ghost imaging in Figure 8a–d [92]. The imaging setup is succinct due to the disposal of additional SLMs. Hyperspectral imaging is demonstrated to take advantage of the broadband modulation. Good spatial and temporal coherence of the modulation makes it possible to retrieve images with enhanced resolution and contrast using the reverse propagation technique. 

ZnTe is also widely used for THz electro-optic detection when the THz pulses and the probe pulses are simultaneously launched to the crystal. Only the THz radiation spatially superimposed with the probe pattern is detectable based on the Pockels effect. The spatial and temporal information of the THz beam can be read out by dynamically changing the probe beam pattern and measuring the polarization rotation using balanced CCD cameras or photodiodes [93]. 

In addition, ferromagnetic heterostructures have been revealed as efficient spintronic THz emitters through the spin-to-charge current conversion under the external magnetic field [95,96,97]. The longitudinal spin current $J_{s}$ in the ferromagnetic film induced by the fs laser oscillator is converted into the tangential charge current $J_{c}$ in the nonferromagnetic film due to the inverse spin Hall effect. The charge current generates THz radiation whose electric field is described by
(4)$$E_{THz}\left(t\right)\propto J_{c}=\gamma J_{s}\times M/\left|M\right|$$
where *γ* is the spin Hall angle and ***M*** is the magnetization. The spintronic THz radiation covers almost the entire THz band with additional merits of high conversion efficiency and low pump power. The spatial modulation of the fs laser via commercially available DMDs imprints the pattern into the THz radiation. Considering the few-nanometer thickness, the modulation pattern in DMDs can be well maintained in the THz beam without diffraction blurring. The structured near field illumination can be used for micrometer- or even submicrometer-scale microscopy in Figure 8e–g [94], with the pixel size far less than the THz wavelength. A recent study on ferromagnetic and antiferromagnetic heterostructures offers a unique way to enable spin-to-charge current conversion without external magnetic field but induced by the exchange-bias effect [98]. The efficient generation and spatial modulation of the spintronic THz radiation can be realized in a magnetic field-free manner in the near future. 

## 8. Discussion of Modulation Specifications

There are still other types of modulation not covered by the above discussion. For example, using the strong absorption in water, Poland et al. proposed a THz SLM based on digital microfluidic array [99]. The distribution and movement of water droplets are electrically controlled to turn on or off the pixels [100]. The recent progress of the photoconductive antenna array proves a promising type of THz emitters with reconfigurable radiation patterns [101] and THz array detectors [102,103] for multichannel spectroscopy. Organic semiconductors are reported as high-speed photoconductive THz SLMs [104]. Metasurfaces made of superconductive materials are also good candidates for electrically controlled spatial modulation with large modulation depth [105]. Ge_2_Sb_2_Te_5_ can be repeatedly switched among amorphous, crystalline and intermediate states with the help of nanosecond laser pulses and thermal annealing, enabling nonvolatile, reconfigurable, multi-level and broadband SLMs [106,107].

The specifications of the THz SLM play a key role in determining the ultimate systematic parameters. We list some representative SLMs of different types in Table 1 and compare key specifications.

(1) Operation bandwidth: The modulators, whose modulation mechanism is due to the change in resonance intensity or the movement of resonance frequency, usually have narrow working bandwidth and enhanced modulation depth. SLMs that combine metasurfaces with tunable materials are narrowband modulators with a bandwidth of no more than hundreds of GHz [32,47]. The modulator that only depends on the change of carrier concentration has a wide bandwidth due to its non-resonant characteristics. Light addressable semiconductors, VO_2_ and diffractive micromirrors have the advantage of wide bandwidth. Spatially modulated spin-to-charge current transition offers an even wider bandwidth, which are very suitable for hyperspectral imaging.

(2) Spatial resolution: The pixel size is the main factor determining the spatial resolution. For electrically controlled modulation, the size of THz metasurface unit is usually tens to hundreds of microns. One electrode is often used to control multiple units to simplify the control circuit. The pixel size is generally greater than 1 mm and the spatial resolution is low. Once the processing is completed, the pixel size is fixed. The resolution of electrically addressable micromirror array can be varied, but it must be integral times of the minimum micromirror size. The pixel size of the optically pumped modulator can be flexibly controlled by DMD and is only limited by the optical diffraction. The spatial resolution is high, and the pixel size can reach 10 μm or smaller [71,94]. 

In addition to the pixel size, the spatial resolution of the structured THz beam is also affected by the crosstalk between pixels and the transmission distance. Because of the diffusion of photoexcited carriers, there is crosstalk between pixels in the optical addressing modulator, while the electrical addressing shows a better confinement and lower crosstalk [61]. As the modulated THz wave propagates in space, the diffraction effect will further reduce the spatial resolution. So, thin-film modulators made of graphene, VO_2_ and ferromagnetic films can better maintain the modulation pattern at the exit interface without apparent blurring. 

(3) Number of pixels: The number of pixels depends on the addressing method. Due to the complex control circuit, electrically addressable SLM is often designed as a 1D modulation device, and the number of pixels reported is less than 100. The number of pixels of optically addressable SLM depends on the number of pixels of DMD, with reported results in the scale of 64 × 64 or 128 × 128 pixels.

(4) Modulation rate: One factor limiting the modulation rate is the response rate of the tunable material. The carrier lifetime of semiconductors is generally in the order of nanoseconds to microseconds [65], so the switching rate of electrically addressable semiconductor SLM can be as high as megahertz, while the modulation rate of optically addressable SLM is limited by DMD and the repetition rate of the fs pulse laser if needed. DMDs usually show the refresh rate of tens of Hz. More advanced high-performance DMD can achieve tens of kHz [67]. The amplified Ti: sapphire fs laser has the repetition rate of kHz, while the repetition rate of fs oscillator is tens of MHz. If high power is not need, the fs oscillator helps to achieve faster modulation with lower cost. The liquid crystal is limited by the response time, and the modulation rate is only tens of Hz. Due to the phase transition of VO_2_ through Joule heat, the slow temperature response limits its modulation rate to less than a few Hz. 

(5) Modulation depth: Micromirrors and the THz emitter array can completely switch the pixels on or off, so a large modulation depth can be obtained. When the semiconductor thickness is greater than the skin depth, nearly 100% modulation depth can be obtained under strong enough optical pump power. However, when the volume limits the number of adjustable carriers in ultrathin seminconductors, VO_2_ and graphene, the modulation depth decreases. When combined with metasurfaces, the large absorption or radiation loss restricts the modulation depth. In addition, the modulation depth is closely related to the modulation rate. With the increase in the modulation rate, the modulation depth decreases gradually. LC SLM is mostly used for spatial phase modulation. The phase modulation range is mostly limited to 0–180°, and the continuous phase modulation covering 360° is still challenging.

## 9. Applications and Challenges

With the ultimate capability of beam manipulation, THz SLMs are a type of developing active components necessary in many different application scenarios. One of the most exciting applications is the computational ghost imaging or simply single-pixel imaging. This process can be mathematically described as projection of the object vector into a new basis with each base vector being the modulation pattern of the THz SLM. The projection coefficients are the time-resolved detection signals. With the predefined modulation patterns and correlated detection signals, the object can be reconstructed through matrix inversion. Detailed theory can be found in related reviews [31,108,109,110,111]. Since the intensity and phase patterns are both reconstructable [112], the imaging system is applicable to various materials, such as metallic resolution chart [71], knifes [65], plastic [14], semitransparent materials [113], leaves [67] and circuit boards [70]. The hyperspectral imaging can further provide the spectral information and composition information of the object. Studies to track hydration changes in a leaf, to observe the photoconductivity distribution of graphene [114] and to monitor the relaxation process of charge carriers undoubtedly prove the importance of THz SLMs and the uniqueness of THz imaging in exploring macro- and micro-scale materials. The state-of-the-art imaging resolution is λ/133 at 0.5 THz by using the optically encoded VO_2_ thin film modulator [77]. In terms of imaging speed, the DMD-encoded silicon total internal reflected modulator enables real-time imaging at six frames/s with 32 × 32 pixels aided by the compressed sensing [67]. Graphene modulator arrays are used for THz imaging with video-rate modulation speed and 16 × 16 pixels [84]. In addition to the compressed sensing technique, the sampling time can be further reduced by deep-learning neural networks while retaining high imaging quality and good signal-to-noise ratio [115,116].

Spatial phase modulation shows better signal-to-noise ratio as the energy is not blocked during modulation. In addition, it offers unprecedented possibility for dynamic wavefront shaping. Different patterns can be imprinted in a silicon film by DMD-programed optical pump, so that the THz wavefront can be tailored for real-time holographic image display, dynamic focusing [35] and generation of vortex beams with tunable topological charges [37]. Spatial phase modulation with variable phase gradient leads to dynamic steering of the THz beam, which will facilitate the development of future 6G wireless networks. The LC-based SLMs can scan the THz beam from 0° to 30° or more [52,53]. The angular steering range of graphene-based SLM reaches 25° [33]. The THz SLMs can behave as intelligent surface to enable information transmission by bypassing different obstacles. They also play vital roles in the communication among different base stations.

Technical challenges of THz SLMs mainly originate from three aspects, the electrically small and optically large feature size, combination of complex metasurface pattern and biasing circuits and the requirement of expensive femtosecond laser. The feature size in tens of and hundreds of micrometers is large for mechanical deformation and LC reorientation, which also poses high requirement to external stimuli. Energy consumption and heat dissipation cannot be neglected. The cost-effective printed circuit board process is not a viable way to fabricate the electrically small structures, which are usually replaced by more expensive ultraviolet lithography. In addition, electrically driven SLMs need to pattern multiple layers with some for resonances and some for electrodes, which further challenges the processing technique. The optical pump using the femtosecond laser oscillators or amplifiers is helpful to inhibit carrier diffusion and indispensable for simultaneous generation and modulation of the THz beam, but is mainly limited to laboratory study due to the cost. 

## 10. Conclusions

In summary, this paper systematically overviews the research progress of THz SLMs categorized from the perspective of functional materials and modulation schemes. Each design has its own advantages and disadvantages. The development of THz SLMs with a number of excellent specifications is critical for practical applications. Higher resolution and faster modulation rate are the major targets to pursue. 

Additionally, researchers are actively exploring new tunable materials and modulation effects. The recent progress of metasurface high-Q resonators governed by the quasi-bound states in the continuum offers a fascinating platform for developing THz SLMs [117,118]. At the same time, the physical mechanism of improving the efficiency [119] and bandwidth [120,121] of metasurfaces in other applications can be used as a guidance to reduce insertion loss and increase the modulation bandwidth of THz SLMs. With the advance of artificial intelligence, the THz SLMs may progress from user-defined modulation into intelligent modulation by judging the excitation condition and automatically setting the controllers [122]. There is no doubt that significant progress has been made in the research area of THz SLMs, and the study is moving towards practical applications in the near future.

## Figures and Tables

**Figure 1 micromachines-13-01637-f001:**
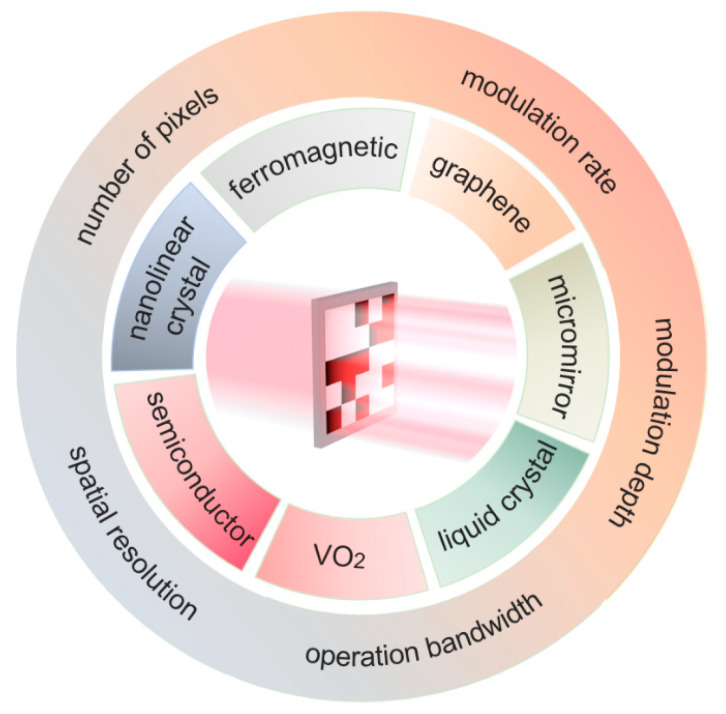
Terahertz spatial light modulators with the functional materials and specifications.

**Figure 2 micromachines-13-01637-f002:**
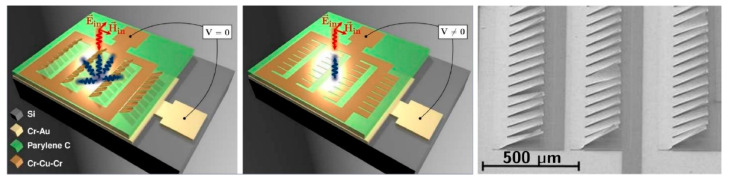
THz micromirror SLM based on self-assembled metal multilayer structure. Reproduced with permission from [44]. Copyright 2019 Springer Nature.

**Figure 3 micromachines-13-01637-f003:**
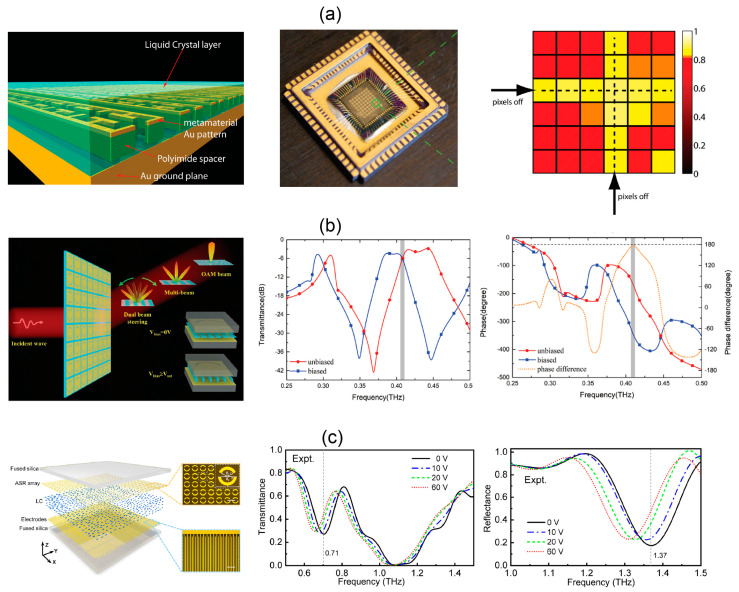
THz metasurface SLMs based on liquid crystals. (**a**) Reflective LC SLM based on a THz metasurface absorber, the packaged sample and a spatial absorption pattern. Reproduced with permission from [51]. Copyright 2014 John Wiley & Sons, Inc. (Hoboken, NJ, USA) (**b**) Transmissive LC−metasurface SLM and its binary phase modulation. Reproduced with permission from [53]. Copyright 2021 John Wiley & Sons, Inc. (**c**) Transflective SLM based of LC sandwiched by asymmetric split rings and interdigital electrodes, which spatially modulate the THz intensity on both sides of the device and at two frequencies according to the incident polarization. Reproduced with permission from [55]. Copyright 2022 Optica Publishing Group.

**Figure 4 micromachines-13-01637-f004:**
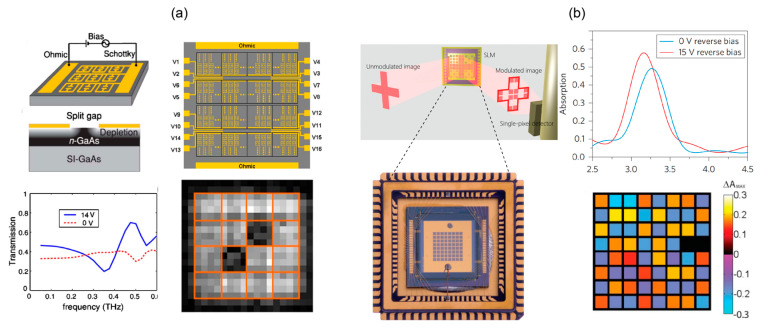
Electrically biased semiconductor THz SLMs. (**a**) Transmissive semiconductor SLM. The geometry of a pixel as a Schottky junction formed by an SRR metamaterial with n-GaAs (top left), the layout of the SLM (top right), the transmission spectra of the pixel with and without electric biasing (bottom left) and a spatial absorption pattern (bottom right). Reproduced with permission from [61]. Copyright 2009 AIP Publishing LLC. (**b**) Reflective semiconductor SLM based on metamaterial absorbers. Modulation of the absorption is done by the bias (top right). Spatial distribution of differential absorption maps (bottom right). Reproduced with permission from [32]. Copyright 2014 Springer Nature.

**Figure 5 micromachines-13-01637-f005:**
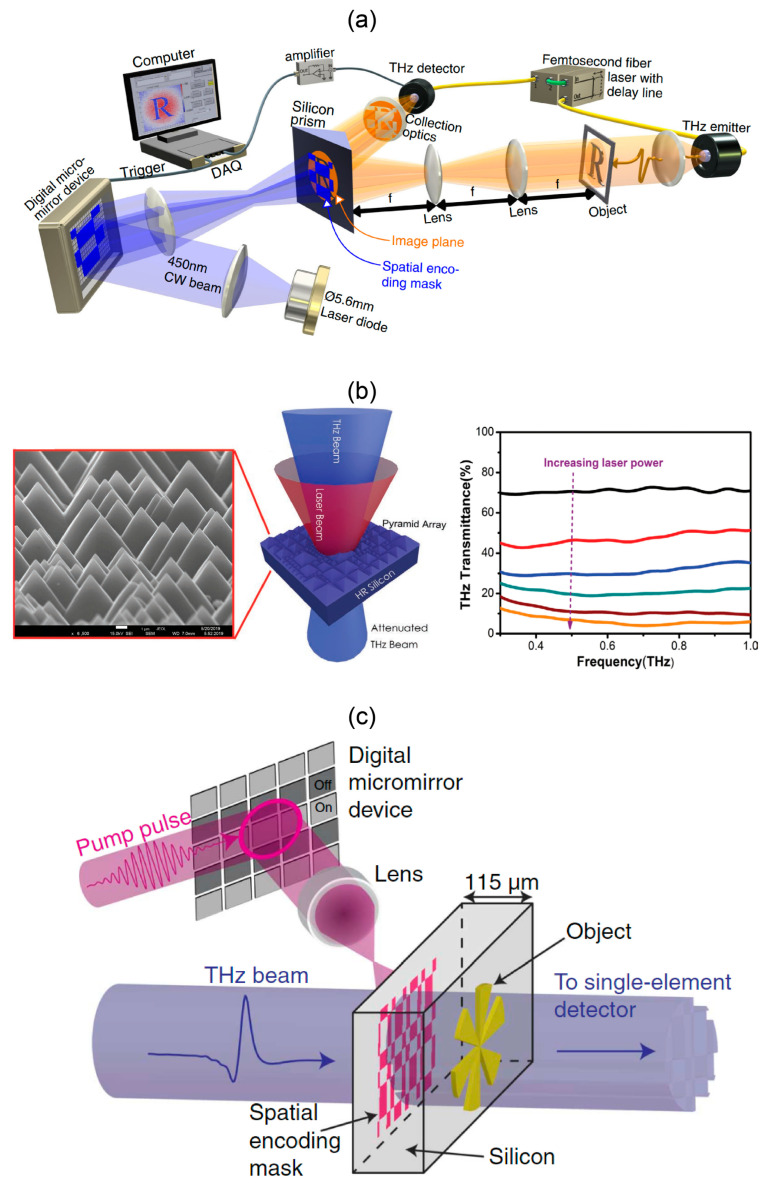
Optically pumped semiconductor THz SLMs. (**a**) Optically pumped silicon prism for total reflection spatial modulation and single pixel imaging. Reproduced with permission from [67]. Copyright 2020 Springer Nature. (**b**) The micropyramid array reduces reflection of the pump laser at the semiconductor interface and increases the modulation depth. Reproduced with permission from [69]. Copyright 2020 John Wiley & Sons, Inc. (**c**) Near field THz SLM for high-resolution imaging. Reproduced with permission from [70]. Copyright 2016 AAAS.

**Figure 6 micromachines-13-01637-f006:**
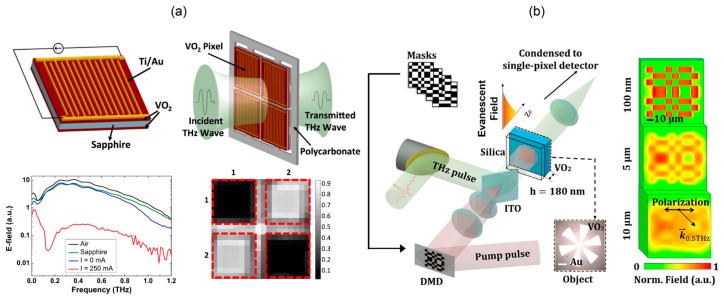
VO_2_-based THz SLMs. (**a**) Pixels made of double-layer VO_2_ and interdigital electrodes, with their voltage-controlled transmittance and measured spatial pattern. Reproduced with permission from [76]. Copyright 2015 Elsevier B.V. (**b**) Femtosecond pumped VO_2_ near-field SLM and the single-pixel imaging system, with the plots on the right showing diffraction effect of the spatially modulated pattern after propagation in the substrate of different thickness. Reproduced with permission from [77]. Copyright 2019 Optica Publishing Group.

**Figure 7 micromachines-13-01637-f007:**
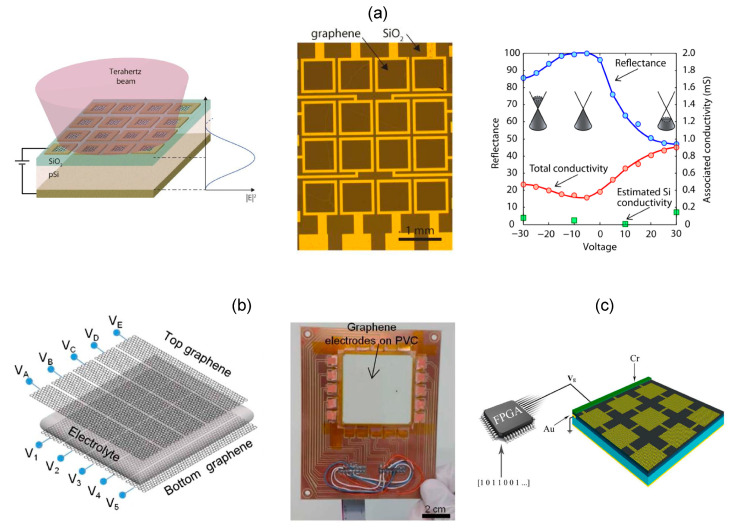
Graphene-based THz SLMs. (**a**) Electric-driven graphene reflective SLM (**left**), the microscopic image (**middle**), and the variation of the reflectivity with the bias voltage (**right**). Reproduced with permission from [81]. Copyright 2013 Optica Publishing Group. (**b**) Graphene SLM based on ionic liquid electrolyte gating method for large modulation depth, the geometry (**left**) and the photo of the device (**right**). Reproduced with permission from [82]. Copyright 2015 Optica Publishing Group. (**c**) Reflective SLM for phase modulation using patterned graphene. Reproduced with permission from [83]. Copyright 2018 Elsevier B.V.

**Figure 8 micromachines-13-01637-f008:**
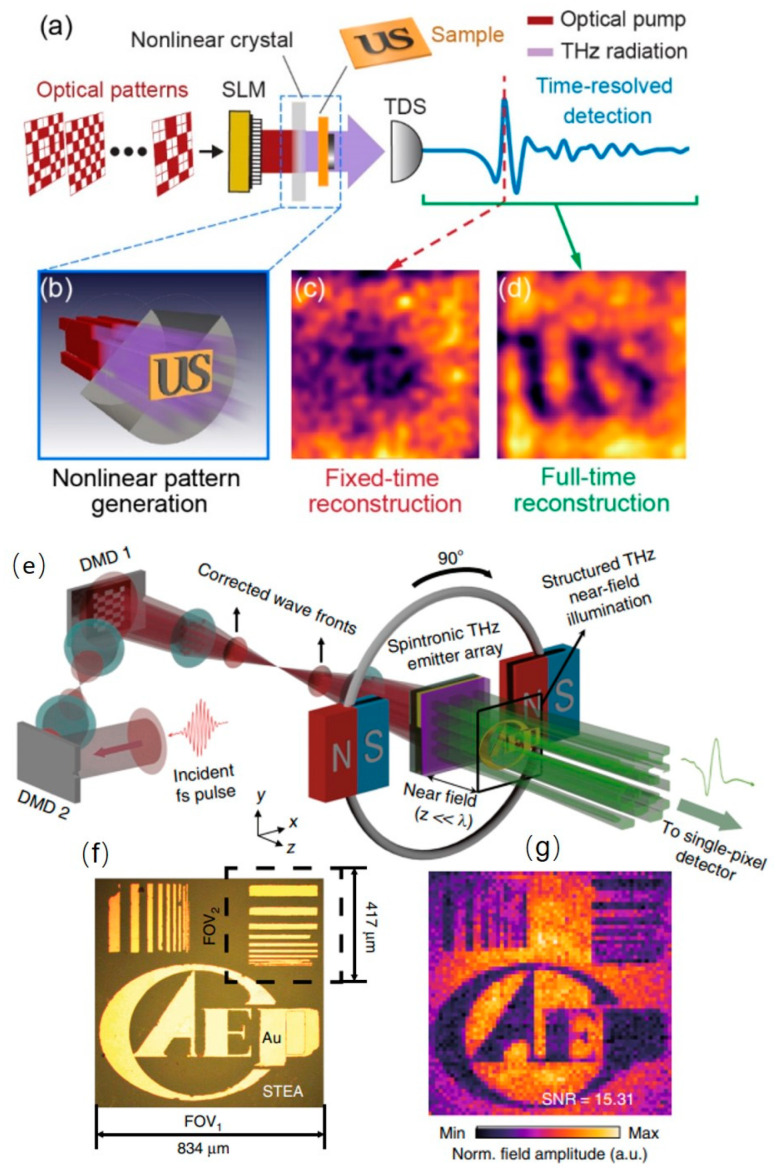
THz spatial modulation during generation. (**a**–**d**) Single-pixel imaging enabled by spatially modulating the pump laser shined onto the nonlinear crystal, which generates THz patterns for image reconstruction. Reproduced with permission from [92]. Copyright 2018 American Chemical Society. (**e**–**g**) The spintronic THz emitter array excited by DMD-encoded fs laser radiates structured THz beam for near field ghost imaging. Reproduced with permission from [94]. Copyright of 2020 Springer Nature.

**Table 1 micromachines-13-01637-t001:** Comparison of the specifications of the representative THz SLMs.

Material	Modulation Method	Band Width	Pixel Size	Pixel Number	Modulation Rate	Modulation Depth	Refs.
micromirror	electric	wide	1 × 2 mm^2^	4 × 6	1 kHz	>50%	[44]
LC	electric	narrow	0.64 × 15.36 mm^2^	1 × 24	-	180°	[53]
LC	electric	narrow	5 × 5 mm^2^	12 × 12	13 Hz	32°	[54]
semiconductor	optical	wide	0.1 × 0.1 mm^2^	128 × 128	1050 Hz	90%	[70]
semiconductor	electric	narrow	0.6 × 0.6 mm^2^	8 × 8	12 MHz	62%	[64]
VO_2_	optical	wide	4.5 × 4.5 μm^2^	64 × 64	A few Hz [75]	30%	[77]
VO_2_	electric	narrow	1900 × 1900 μm^2^	8 × 8	1 kHz	65%	[78]
graphene	electric	narrow	0.7 × 0.7 mm^2^	4 × 4	6 kHz	50%~30%	[81]
spintronic heterostructure	optical	wide	6.5 × 6.5 μm^2^	128 × 128	1 kHz	-	[94]

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
