# Peer review of "Recent Progress of Terahertz Spatial Light Modulators: Materials, Principles and Applications"

_micromachines, 2022, doi:10.3390/mi13101637_

Round 1
Reviewer 1 Report
This paper reported different materials and principles for terahertz spatical light modulation. The topic is very interesting. And I think it would be attractive for the readers. Moreover, the manuscipt was organized well and it could accepted for publication.
Reviewer 2 Report
Thank you for delivering a nice manuscript related to terahertz spatial light modulators. The writing and presentation quality of this work is nice. This review presents the recent progress of THz spatial light modulators from the perspective of functional materials, and analyzes their modulation principles, specifications, applications and possible challenges. The authors must provide comprehensive Tables and all figures must be provided in high resolution. A few suggestions are
1. Please add an acronym Table to define all abbreviations.
2. Please clearly indicate the organization of this study.
3. It will be interesting to add a comparative Table in the Introduction section and compare your work with existing reviews or surveys.
4. It is also suggested to provide a precise summary in the start of each section to define contributions in it.
5. Figures quality must be improved. The inside text is blur in some figures. All figures must be provided in high resolution for better readability.
6. There is a complete lack of discussion and references in Section 2.
7. I suggest authors must add a comprehensive Table in each to precisely summarize the discussion within Tables Readers can fastly understand research contributions from Tables.
8. Again, there is a lack of reference literature in Section 5.
9. In Table 1, the authors have ignores several reported studies in this domain. Please revise and add more research contributions. Specifically, add research works reported in 2021-2022.
10. It will be interesting if the authors can add a few technical challenges in a separate section.
11. The authors must provide a separate section for Conclusion. It must be precise and add a few possible future aspects.
In my opinion, the overall research contributions are good and satisfying to be published after these minor revisions.
Reviewer 3 Report
This manuscript overviews the research progress of THz SLMs from the perspective of functional materials and modulation schemes. The content is well organized and presented. This manuscript can be published by the current format
